# Psycho-Social Correlates of Cyberbullying among Polish Adolescents

**DOI:** 10.3390/ijerph20085521

**Published:** 2023-04-14

**Authors:** Sławomir Rębisz, Aleksandra Jasińska-Maciążek, Paweł Grygiel, Roman Dolata

**Affiliations:** 1Institute of Education, University of Rzeszów, ul. Ks. Jałowego 24, 35-959 Rzeszów, Poland; 2Faculty of Education, University of Warsaw, ul. Mokotowska 16/20, 00-561 Warsaw, Poland; 3Institute of Education, Jagiellonian University, ul. Batorego 12, 31-135 Kraków, Poland

**Keywords:** cyberbullying, cybervictimization, adolescents, self-regulation, social skills, peer relationships, problematic Internet use, two-part regression model

## Abstract

Cyberbullying has recently attracted attention due to its increasing prevalence and serious consequences for both victims and perpetrators. The objective of this population-based study was to examine the determinants of a person becoming a perpetrator of cyberbullying, including personal resources (emotional self-regulation, self-esteem, internal locus of control, optimism), social skills (prosocial behavior, assertiveness, cognitive empathy, cooperation), peer relationships (peer support, threats from peers, peer rejection, dislike of peers), and problematic Internet use (excessive Internet use, impulsive reactions to Internet deprivation). Participants (N = 541) were students at elementary schools (age 14–15) in Ostroleka, a city in central-eastern Poland. Two-part regression was used to explore protective/risk factors of the likelihood of an individual using cyberviolence (dichotomous part: involvement in violence) and how often it is used (continuous part: frequency of cyberbullying). The results showed that the emotional component is crucial to cyberbullying, as indicated by the importance of emotional self-control, which reduces the frequency of cyberbullying. Other important factors are assertiveness, impulsive response to limited Internet access (which increases the likelihood of engaging in cyberbullying) and fear of peers (which reduces its frequency). In turn, the importance of pro-sociality (which inhibits engagement) and peer support (which promotes engagement) points to the second important component of cyberbullying—that is, group mechanisms. At the same time, the results indicate that while the importance of Internet addiction as a risk factor for cyberbullying should not be underestimated, the amount of time spent online cannot be seen as the source of the problem. The study leads to the conclusion that effective interventions targeting cyberbullying should focus on the development of more adaptive styles of coping with emotions.

## 1. Introduction

In contrast to previous generations, today’s youth are growing up in a culture in which being online is an integral part of everyday life [1]. Although the development of technology opens up a world of opportunities, it also carries the danger of adverse phenomena, such as cyberbullying [2], which is experienced by between 10% and 53% of young people, depending on the various definitions of cyberbullying and the age of respondents in the groups studied [3].

### 1.1. Bullying and Cyberbullying

Bullying is commonly defined as repeated exposure to negative behaviors of one or more peers that are intended to cause harm and involve a power imbalance in favor of the perpetrator [4]. Due to its nature (physical, verbal, or relational), bullying behavior is associated with negative psychological outcomes for both the perpetrator and the victim [5,6].

Cyberbullying, in turn, is defined as being cruel to others by sending or posting harmful material or engaging in other forms of social cruelty while using the Internet or other digital media. Cyberbullying shares many characteristics with “traditional” violence, such as intentional harm, repetitive engagement, and power imbalances [7]. Unsurprisingly, the existing meta-analyses show positive associations between involvement in traditional bullying and cyberbullying [8,9].

However, as Kowalski [8] notes, despite their co-occurrence, the two behaviors differ in important ways. *First*, cyberbullies are often anonymous, which significantly expands the pool of potential perpetrators of cyberbullying compared to the traditional type. *Second*, the difference lies also in the accessibility of the victim. Traditional bullying by children or adolescents occurs most often at a particular educational institution during the school day, whereas the perpetrators of cyberbullying can be active via electronic devices round the clock and seven days a week. *Third*, as a result of the use of networks and electronic devices for cyberbullying, the potential audience for violent behavior is much larger than for traditional bullying. *Fourth*, those who engage in cyberbullying are not exposed to the direct effects their actions have on the victim. *Fifth*, due to the nature of the technology used for cyberbullying, the victim’s response may be delayed [8]. *Sixth*, cyberbullying is, on the whole, more indirect and involves minimal physical force [10]. *Seventh*, whereas traditional bullying consists mainly in physical, verbal, and social forms of violence, cyberbullying takes visual and textual forms [11].

### 1.2. Cyberbullying—Age and Gender

Research findings on the age determinants of cyberbullying are inconclusive [12]. Some studies suggest that it is more prevalent in middle schools [13], while others suggest that older students (15+) are more likely to cyberbully than younger ones (10–14 years old) [14,15]; also, it seems that involvement in cyberbullying increases in high school, while other forms of bullying peak in late elementary or middle school and decline in high school [16,17]. Some reviews suggest that engagement in cyberbullying peaks during adolescence. Accordingly, the experience of victimization of cyberbullying peaks among 12–14 year-olds [18].

Various studies demonstrate that when it comes to traditional bullying, boys are more likely to engage in it than girls, particularly in regard to direct physical abuse [10]. Many studies have found that men are also more likely to participate in cyberbullying than women [6,10,17]. One meta-analysis [19] concluded that the main reason adolescents engaged in cyberbullying was relationship problems, and that girls fell victim to it in most cases. Research conducted among students showed that girls were more likely to report being victims of relational and verbal cyberbullying, while boys were more likely to report physical harassment [17].

Although many previous studies indicate that males are more likely to engage in cyberbullying than females, there is now evidence that perpetration and victimization are equal in both genders [12,20,21]. However, research indicates that gender differences in cyberbullying behavior vary by geographical region; hence, as Barlett and Coyne [22] note, cyberbullying should be conceptualized as a social and cultural phenomenon [10].

### 1.3. Cyber Perpetrator vs. Cyber Victim: The Cyclic Process Model

Many studies show that a lot of teenagers who are targets of aggression become perpetrators of online violence. This mechanism is explained by the *Cyclic Process Model* theory, which assumes that face-to-face victimization in teens triggers feelings of anger and frustration, and individuals who experience anger and frustration are more likely to engage in deviant behavior. Increased levels of tension in bullied teens may thus find an outlet in cyberbullying behavior [18]. Additionally, a meta-analysis conducted by Guo [3] found that the variables related to prior experiences of being cyberbullied or exposed to traditional bullying are the two strongest predictors of engaging in cyberbullying.

### 1.4. Personal Resources and Cyberbullying

Of the constructs describing characteristics of students, the following have been chosen for consideration in this article: emotional self-regulation, self-esteem, internal locus of control, and optimism.

Contemporary approaches to self-regulation stem from the work of Albert Bandura [23], who argued that, according to the theory of social cognition, human behavior is motivated and regulated by the constant exertion of self-influence. This influence concerns the self-control of behavior and its social context, evaluation of one’s own behavior in relation to personal standards and environmental circumstances, and affective reactions. Even though there are many other theoretical approaches to self-regulation, it is usually defined as the ability to control one’s behavior, emotions, thoughts, and attention in order to achieve goals [24]. 

As already mentioned, many studies expose the disturbing fact that adolescents who are objects of aggression become the perpetrators of online violence [9,18]. One way to explain this is that feelings of anger and frustration find expression in cyberbullying others [25]. It seems that this self-perpetuating cycle of violence may disrupt an individual’s ability to self-regulate, especially emotionally. Since research shows that anger mediates the relationship between being a victim and carrying out acts of cyberbullying [26], it is the self-control of negative emotions that may be the key to breaking the cycle of aggression.

A study by A. Baroncelli and E. Ciucci [27] showed that the emotional intelligence dimension of emotional regulation (that is, a factor strongly related to emotional self-regulation) was linked negatively to both traditional violence and cyberbullying. However, the relationship disappeared when traditional bullying was controlled for, while it continued when cyberbullying was controlled for offline violence and being a victim. This means that self-regulation is probably more relevant to cyberbullying than to offline aggression.

Cyberbullying can result from a lack of sufficient emotional self-regulatory capacity, but the picture is even more complicated by the fact that aggression, especially in its instrumental form, can itself be a manifestation of self-regulation. For example, in individuals with high-level narcissism, aggression may be an attempt to regain social status after their social reputation has been threatened [28]. Studies on the relationship between self-regulation and cyberbullying also emphasize the complexity of the relationship; for example, A.H. den Hamer and E.A Konijn [29] found in their longitudinal study that using positive emotional regulation strategies to cope with anger did not reduce cyberbullying behavior, although, on the other hand, adolescents who blamed others and ruminated on their negative experiences were more likely to engage in cyberbullying.

Thus, presenting self-regulation as a simple remedy for cyberbullying has no clear empirical grounds. Certainly, the inhibitory effect of self-regulation is worth exploring further, especially while considering multiple potential causal factors.

Another construct under consideration in this context is general self-esteem, understood as an overall and subjective self-evaluation. Self-esteem is a dynamic phenomenon; studies show that it undergoes particularly big fluctuations during childhood, adolescence, and old age [30]. The level of self-esteem of boys and girls in late childhood (between 7 and 10–12 years of age) is similar. Differences become apparent only in adolescence (between 10–12 and 20–22 years of age), especially in its initial phase. This implies the need to take gender into account when analyzing the correlates of self-esteem.

The associations of self-esteem with cyberbullying relate primarily to being a victim. Self-esteem, including suicidal thoughts and suicidal behavior, has been found to be significantly associated with being a victim of cyberbullying [31]. However, a study of Australian elementary school students between the ages of 10 and 12 showed a more complex picture. Children who reported a single episode of cyberbullying had the highest levels of self-esteem, while those who reported multiple acts of cyberbullying had the lowest [32].

The impact of being a victim of online violence on self-esteem is well documented empirically, but does the inverse relationship also hold? It is known that being a victim and perpetrator of online aggression are strongly, positively related (Cyclic Process Model). Is it to be expected, then, that low self-esteem fosters cyberbullying?

Research usually points to a negative relationship, but the picture is far from clear. The results of some research reviews [33] show that low self-esteem is associated with the risk of both cyberbullying and cybervictimization, as confirmed by a 2018 meta-analysis [34]. However, the authors themselves point out as a limitation that it is impossible to say whether high self-esteem protects against bullying, or bullying is the cause of low self-esteem. One of the more recent meta-analyses using the results of longitudinal studies has provided stronger evidence, confirming a negative association of being a victim with self-esteem, but not confirming the correlation of self-esteem with being a perpetrator of cyberbullying [35].

In fact, a study of students in grades 6–12 in the US [31] provides even more complex results. The participants were categorized as belonging to one of four groups: victims, perpetrators, both perpetrators and victims, and those not involved in violence at all. Students in the victim/perpetrator group, both online and offline, had the most negative scores on most measures of mental and physical health, and academic performance. Self-esteem was also at its lowest in this group. The most positive results were found in the group completely uninvolved in violence. Surprisingly, however, no differences were recorded in the groups of ‘perpetrators only’ and ‘victims only’ and students who belonged to them had higher levels of self-esteem than the victim/perpetrator group (but significantly lower than the group not involved in any kind of violence). The results for traditional and online violence were similar. This is an unexpected outcome, as one would presume that the students who were only victims of cyberbullying would have the lowest self-esteem.

A recent study of the relationship of interest, conducted in Italy on a group of 13–16-year-old students, also tells a complex story [36]. Students classified as belonging to a group with a self-derogation profile were more exposed to being both a victim and perpetrator of violence, online and offline equally. In contrast, high body- and peer-oriented self-esteem significantly increased the risk of engaging in problem behaviors such as mobbing. Finally, students with consistently high self-esteem profiles were more protected from bullying/cyberbullying and victimization/cybervictimization behaviors compared to students with a self-destructive profile.

As in the case of self-regulation, research does not provide a clear answer to the question of the impact of self-esteem on cyberbullying. While the relationship between being a victim of cyberbullying and having low self-esteem seems to be well confirmed [8,37], the influence of self-esteem on cyberbullying is not certain, although is likely.

Another predictor of cyberbullying is internal locus of control. The construct of locus of control was introduced into psychology in the 1950s by Julian B. Rotter [23] and is conceptualized as a generalized expectation of internal or external reinforcement control. In other words, it refers to people’s beliefs about where factors influencing events in their lives are located (i.e., inside or outside the subject) and whether they can be controlled.

Some studies have shown that people develop more internal locus of control with age, but it seems more appropriate to treat internal and external loci of control as orthogonal dimensions with different developmental trajectories [38]. Thus, internal locus of control increases until middle age, while external locus of control first weakens and then again increases in adulthood.

Internal and external locus of control are predictors of many behavioral, cognitive, and affective characteristics of an individual, including psychological well-being or job satisfaction [39]. However, no meta-analysis on protective factors against cyberbullying has found that locus of control matters [34,35]. Only isolated reports indicate the existence of a relationship. For example, a study involving a group of Turkish adolescents found that external locus of control was significantly associated with both being a victim and perpetrator of cyberbullying [40]. In contrast, in a Chinese population of elementary school students, locus of control was proven to moderate the relationship between victimization and the developmental trajectories of internalizing and externalizing problems. Children who experienced higher levels of victimization and exhibited a concomitant external locus of control were more likely to experience internalizing and externalizing problems [41]. These unclear empirical correlations suggest that research into the relationship between locus of control and cyberbullying should be treated as exploratory.

The last of the constructs analyzed in terms of students’ personal resources is optimism. In the 1990s, hope began to be viewed as a one-dimensional construct involving the perception that certain goals could be achieved [42]. Dispositional optimism and pessimism are defined as personality traits characterized by a tendency to expect positive or negative outcomes in life [43]. Optimism has been linked to a wide range of measures of psychological well-being and health [44].

Few studies have directly examined any association of optimism as a personality disposition with cyberbullying. One of them is a study conducted on a large sample of Spanish 10- to 12-year-old students, which explored whether victim and perpetrator roles in cyberbullying are related to dispositional optimism [45]. The results showed that optimism correlates negatively with violence when it came to being a victim and a perpetrator.

While emotional self-regulation can break the cycle of aggression by controlling negative emotions, it seems that optimism, with its expectation of a positive resolution of conflict, can be a factor in reducing the likelihood of aggressive responses. An interesting study indirectly illuminating this hypothesis was conducted on a group of Israeli adolescents [46], revealing that the extent of exposure to armed conflict via the media was directly related to psychological distress and post-traumatic symptoms. Dispositional optimism moderated the relationship between traditional media exposure to war images and psychological distress. However, in the case of social media exposure, none of the factors—including optimism—were protective against negative effects.

The significance of optimism in countering cyberbullying is indirectly confirmed by interventions aimed at combating traditional bullying which focused on various aspects of well-being, and also on optimism; they have yielded positive results in terms of improved coping with conflict [47]. Although the theoretical rationale and a small number of studies [8] suggest a negative association between optimism and being a perpetrator of cyberbullying, it is still an area of great uncertainty.

### 1.5. Social Skills and Cyberbullying

Social skills are a multidimensional construct that refer to the ability to integrate cognition, affect, motivation, and behavior to succeed in social tasks and achieve positive developmental outcomes [48]. They are an essential tool for appropriate human development and the formation of satisfying interpersonal relationships [2]. They include several related inter- and intrapersonal skills that can be organized into sub-dimensions, such as a person’s ability to interact, cooperate, and communicate effectively with others, but also the ability to be assertive, empathetic, and responsible [48].

According to research, violence is linked to the level of such social skills as empathy, assertive behavior, cooperation, and responsibility [49,50,51]. Low social skills can cause a person to be selfish, indifferent, and lacking in empathy. This can give rise to various social problems, such as aggressive behavior, coercion of will, destruction, intergroup conflict, and fighting [52]. In fact, studies by Barlinska, Lalak and Szuster, [53] Martínez, Rodríguez-Hidalgo, and Zych [2] suggest that having well-developed social skills is protective against bullying, including cyberbullying and victimization.

In broad terms, empathy is primarily an affective reaction in response to the situation of another, encompassing the ability to understand and vicariously experience their feelings [54]. A distinction can thus be made between affective empathy (the ability to perceive and feel the emotions of others) and cognitive empathy (the ability to understand and anticipate the emotions and mental states of others by taking their perspective) [53].

Although research suggests that empathy may play an important role in cyberbullying [55], the results of studies on the relationship between the two are inconclusive. Martínez et al. [2], citing the results of meta-analyses, indicate that high levels of affective and cognitive empathy are usually protective factors, while low levels of empathy are a risk factor for cyberbullying [56,57,58,59]. The results of the study by these researchers showed, however, that being a cyberbully was associated with higher affective empathy. These inconsistent results can be explained in several ways—for example, it is possible that perpetrators of cyberbullying understand, and even share, the feelings of others but continue to hurt victims because it is related to their psychosocial profile. They use their high affective empathy to hurt and subjugate their peers, as part of their Machiavellian and manipulative character [2].

Assertiveness is another important social skill, often discussed in this context, defined as one’s ability to express opinions, feelings, and attitudes openly within certain limits. It contributes to self-confidence, and better interpersonal communication and well-being [60,61].

Assertiveness and self-esteem are important factors in adolescence, playing significant roles in relationships and social interactions. Thus, assertiveness is a life skill that, if mastered in adolescence, can help increase awareness of personal rights and distinguish between assertiveness, submissiveness, and aggression. Research indicates that individuals who exhibit assertive behavior tend to have higher self-esteem and are more successful in life [61]. Some findings also suggest that being a victim of bullying (a cybervictim) is unequivocally associated with low assertiveness. In contrast, being a perpetrator of violence, including cyberbullying, is predicted by high assertiveness [2].

Cooperation is the joint effort of individuals intended to achieve a specific goal. It occurs when two or more people work together to solve a problem or create something that cannot be done by one person alone [49]. According to Man, Liu, and Xue [62], student cooperation can result in a variety of outcomes, often being related to the extent of pro-social behavior.

Pro-social behavior, on the other hand, is behavior intended to benefit another person or group of people; it is the ability to share things, help, and support others. The basic condition for the emergence of pro-social behavior is the appropriate interpretation of a situation, i.e., seeing another person’s need (empathy) and recognizing that it can be met [63,64]. Empathy is often considered as an important motivating factor for prosocial behavior [65,66]. A study by Jenkins, Demaray, Fredrick, and Summers [67] found a positive relationship between prosocial behavior and adolescents’ ability to cooperate. In contrast, several other studies have pointed out that, for example, a school climate in which competition and rivalry dominate provokes and escalates aggressive behavior in students. In such an environment, violent behavior, including cyberbullying, is more common [68,69]. As Nasaescu, Zych, Ortega-Ruiz, Farrington, and Llorent [70] state, bullying and cyberbullying tend to create patterns of antisocial behavior. Thus, a cooperative environment and friendly interpersonal relationships are factors needed to effectively reduce violent behaviors, including cyberbullying [34,62].

Considering the above considerations, it is possible to hypothesize that anti-social behavior and low levels of cooperativeness, in addition to low levels of empathy and high assertiveness, will be predictors of cyberbullying behavior. 

### 1.6. Peer Relationships and Cyberbullying

During adolescence, the ongoing separation-individuation process [28,71] results in the relocation of important attachments from parents to peers [72,73]. Supportive peer relationships serve a protective function for an individual and contribute to a number of short- and long-term benefits, promoting the development of complex socio-emotional, cognitive, and educational skills see review by [74]. In contrast, a lack of satisfying peer relationships acts as a stressor [75], leading to long-term negative academic and psychopathological consequences (for review, see [73]).

Empirical studies show, for example, that supportive peer relationships significantly reduce the risk of victimization among adolescents for review see [76]. The meta-analysis conducted by Mitic et al. [74] estimates this effect at −0.27. Moreover, the protective functions of positive peer relationships relate not only to offline but also to online victimization, although in the latter case the effect of the influence is smaller and estimated at −0.14 [74].

Although positive peer relationships reduce the chances of a person becoming a victim, they do not prevent someone from becoming a perpetrator [76]. Research suggests that perpetrators are not socially isolated and do not feel isolated [77]. Bullies tend to affiliate with other aggressive adolescents [78], have reciprocal friendships, mostly among other bullies [79], and receive affection from other bullies [80]. Therefore, it can be reasonably expected that they may not feel they lack peer support. Such conclusions are supported by the results of other studies [81,82]. However, results from previous research indicate that poor perceived peer support increases the likelihood of involvement in the perpetration of cyberbullying [83,84,85,86,87]. Given the mixed results of previous research on the role of peer support in online bullying, we do not hypothesize about the relationship between the two phenomena.

As previous research indicates that a perception of the quality of peer relationships is associated with lower levels of being liked by others, although these relationships are not strong [88,89], the fact that perpetrators are more often disliked may lead to their worse perception of the quality of their peer relationships. Correlation studies suggest that cyberbullies perceive more negative peer relationships among classmates than young people not involved in cyberbullying. For example, they more often reported that students at their school have difficulty getting along, were mean to one another, and picked on other students [90]. Additionally, experimental studies found that teenagers exposed to rejection and manipulation responded more aggressively than others who were included [91,92], and that negative perceptions of one’s own peer status promoted increased levels of aggression in response to negative feedback from a peer in a controlled laboratory situation [93]. In this light, it can be expected that negative perceptions of peer relationships, indicators of which could be the level of perceived rejection by peers or the level of being disliked by them, will promote active cyberbullying behaviour.

Meta-analyses indicate that bullying victimization and perpetration correlate strongly and that the experience of being bullied by others—both offline and online—is a risk factor for the subsequent perpetration of bullying [9]. It is suggested that previous victimization activates a mechanism causing an individual to tend violate social norms [94]. Another mechanism contributing to the cycle of violence [95] may be motivation for revenge [25,96]. Research shows that individuals who are victims “are hypervigilant of potential threats from peers and interpret benign or innocuous stimuli as a threat, thereby increasing state-level anxiety” [97]. To minimize such anxiety, young people may tend to engage in aggressive behaviour to protect themselves from additional victimization. In general, it can therefore be expected that a perceived threat from peers can intensify the tendency to be a cyberbully.

### 1.7. Problematic Internet Use and Cyberbullying 

Cyberbullying takes place through the various means of information and communication technologies (ICT), so the way we use them provides a context for the analysis of the phenomenon [21]. ICT have become an integral part of our lives, facilitating the acquisition of data, communication, work, study, and entertainment. However, the boundary beyond which we are exposed to their problematic use is increasingly being crossed [98,99,100].

Problematic Internet Use (PIU; otherwise known as Internet Addiction) has been the subject of much research, although researchers neither agree on a single definition nor on the diagnostic criteria. The World Health Organization’s International Classification of Diseases, in its 11th edition, included only Internet gaming disorder as a new condition. However, it did not address the broader concept of Internet addiction [101], even though it is a multifaceted problem that goes beyond online gaming abuse and encompasses a variety of online behaviors that impair normal functioning [102,103]. Compulsive thinking about the Internet (Internet-related activities), excessive use of its resources, inability to control the time spent online, and neglecting other responsibilities due to Internet use are taken as hallmarks of PIU [104]. The symptoms are in fact similar to those of psychoactive substance addiction, including withdrawal symptoms, unpredictable behavior, or mood swings [98].

According to a recent systematic literature review, Internet addicts are more likely to be perpetrators or victims of violence and cyberbullying [5]. All the studies described in the referenced article, involving adolescents between the ages of 11 and 22 from ten countries and three continents, found a positive correlation between PIU and cyberbullying (11 studies, two of which involved only victims of cyberbullying) and/or school violence (5 studies). A meta-analysis by Kowalski, Giumetti, Schroeder, and Lattaner [8] estimated the effect of frequency of Internet use on the likelihood of perpetrating cyberbullying to be 0.20, and slightly lower (0.17) for cyberbullying victimization. However, most of the reports were from cross-sectional studies. Longitudinal studies verifying possible mechanisms of influence may help determine the direction of the relationship.

Some researchers [21], in their attempt to explain the mechanism underlying the relationship between PIU and cyberbullying, suggest that excessive screen time increases one’s chances to interact with violent media content, which, according to social learning theory [23], can result in increased aggressive behavior. The claims of this theory in relation to cyberbullying are supported by contemporary research. For example, a large longitudinal study involving Cypriot adolescents found that exposure to media violence is a risk factor for cyberbullying perpetration and victimization [81]. The correlations obtained in this study were not strong, but, importantly, they were determined when controlling for prior levels of violence and cyberbullying and other control variables. The association of exposure to media presenting content including antisocial and risky behavior with the use of cyberbullying was also confirmed in a Dutch longitudinal study involving adolescents aged 11–17 [105]. This study additionally found that high exposure to violence in the media translated significantly higher into incidences of cyberbullying among boys than girls, and among those with initially high levels of interaction with violent media content.

However, it is not always that more time spent online is linked to higher exposure to violent media content. Therefore, alternative explanations of the relationship between time spent online and cyberbullying have been proposed, which emphasize the relevance of prior levels of aggression as an explanatory variable [106], suggesting that individuals with higher levels of aggression spend more time online because it facilitates exposure to aggressive content and engagement in aggressive behavior. A longitudinal study of Canadian adolescents found that, when controlling for prior levels of cyberbullying, frequency of Internet use was not a significant predictor of the perpetration of cyberbullying [107]. Similarly, a longitudinal study of German students aged 9–17 showed that frequency of Internet use was not a risk factor for committing and experiencing cyberbullying, but may rather be a consequence of these [106]. On the other hand, a longitudinal study of the problematic use of the Internet found that prior PIU predicts an increase in the committing of cyberbullying, but that prior cyberbullying does not predict an increase in PIU [108]. This means that it is likely that it is not the amount of time spent online that is the risk factor for an increase in cyberbullying, but rather the loss of control over when and how digital technologies are used.

Another hypothesis of interest in this context is that Internet addiction leads to poorer psychosocial functioning, translating into increased irritability, which can result in aggression [6,21]. Keles, McCrae, and Grealish [109], who performed a systematic review of studies, found that time spent on social media, active use of social media and social media addiction are significant risk factors for depression, anxiety, and psychological distress. Additionally, while these authors conclude that the problem is complex, they also stress that a link between excessive social media use and mental health problems exists without a doubt. Other studies also show a link between PIU and depression [110,111,112], substance abuse [110,113,114], and higher impulsivity and aggression [112,115,116].

In light of the above findings, it seems reasonable to hypothesize that PIU, both in the dimension of excessive use and irritability linked to Internet withdrawal, will be positively correlated with the use of cyberbullying.

### 1.8. Research Objective and Hypotheses

The article focuses on the perpetration of cyberbullying, and the primary objective of the study was to establish the determinants of a young person becoming a perpetrator of cyberbullying. This is an important area of inquiry because, as shown in the literature review, the empirical findings to date for some personal characteristics (self-regulation, self-esteem, empathy, sense of peer group support) are inconclusive, and for others (locus of control, optimism) there is far too little research data. Multivariate studies would seem to be particularly important in this context.

An important contribution of the study presented here is the use of a new approach to operationalize cyberbullying. Studies of cyberbullying lack a convincing standard in this regard. Some researchers operationalize cyberbullying with a single global question, although most use indicators that relate to multiple questions (a summary of studies from this angle can be found, for example, in [4,8]). As Kowalski [8] notes, single-question measurement of cyberbullying, as opposed to multi-question measurement, has a number of serious limitations, such as the rather lower, though incalculable, reliability of the measurement; the inability to take into account the multifaceted nature of the trait under scrutiny and test its structure, and the fact that single-question indicators give lower estimates of the level of cyberbullying, as people are less likely to give honest answers to a global assessment heavily weighted with the label of ‘perpetrator’.

The studies also differ in terms of the reference period used in the questions for the frequency of cyberbullying incidents (e.g., last 6 months, last school year). Since the questions are retrospective in nature, a short period of time favors the quality of recollection; on the other hand, a longer period reduces uncertainty related to possible changes over time in the severity of cyberbullying.

Extremely important for the operationalization of cyberbullying is the determination of the criteria for categorizing the respondent as a perpetrator. Often a classification approach is used (cyberbullying index as a nominal variable), which involves the problem of determining the cutoff point at which someone is assumed to be a perpetrator of cyberbullying [4]. A significant problem with dividing respondents into perpetrators and non-perpetrators using a cutoff point is that its determination is highly arbitrary [117]. In addition, the problem with such an approach is that it loses sight of much of the variance in respondents’ responses, especially when measuring cyberbullying in a multivariate manner.

Because of the well-documented, strong relationship between being a perpetrator and being a victim, a challenge for operationalizing the phenomenon is to use a model for analysis that helps separate the variance in the variable of being a perpetrator of cyberbullying, mediated by being a victim, from the variance specific to being a perpetrator. Past studies have typically addressed this problem by creating a nominal variable that takes into account both aspects of cyberbullying [31,32,118,119]. This procedure is also loaded with a reduction in the variance of the explained variable.

Another important methodological problem is the discontinuity of the phenomenon under study. Cyberbullying studies using a multivariate approach typically show that the distribution of raw or scaled measure results is far from normal and is clearly bivariate; a sizable group of respondents declare no involvement in cyberbullying, while the remainder differ significantly in the frequency of their involvement in cyberbullying. The classification approach does not fully capture the diversity of the latter group.

Given the limitations of previous research, this study considers a very wide range of characteristics so as to control for possible mediating effects and spurious correlations. Predictors of cyberbullying perpetration are analyzed while controlling for being a victim of cyberbullying and for the gender variable. A total of 14 independent variables (in addition to gender and being a cybervictim) are included in the analyses, looking at the determinants of cyberbullying from four different perspectives: personal resources, social skills, peer relationships, and problematic Internet use. Few studies on cyberbullying have simultaneously considered such a wide range of variables.

Cyberbullying was measured using a multi-item scale, the reliability and structure of which were verified. In addition, the variability of the trait was not reduced to categories, but two-part regression modeling was used, which allows for taking into account both the incidence of the phenomenon (participated or did not participate in cyberbullying) and the frequency of its occurrence (severity of the phenomenon). This helped us determine which variables were particularly relevant to becoming involved in cyberbullying, and also which were responsible for the frequency of violent behavior. To the best of our knowledge, this is the first study to propose such a model for analyzing the determinants of cyberbullying.

The study is mainly exploratory, but in several cases prior research has helped us formulate the hypotheses.

It is difficult to make strong hypotheses when it comes to personal resources. In the case of emotional self-regulation and self-esteem, arguments in favor of a negative correlation with perpetration of cyberbullying dominate, and one can risk the hypothesis that the two characteristics are protective factors (H1, H2). However, in the case of locus of control and optimism, there are too few arguments to put forward a hypothesis. Therefore, a research question was raised about whether these variables are significant predictors of cyberbullying at all (Q1, Q2).

In the case of the social skills investigated in the study, the state of knowledge to date allows us to formulate the hypothesis that anti-social behavior and low levels of cooperativeness, in addition to low-level cognitive empathy and high assertiveness, will act as predictors of cyberbullying behavior (H3, H4, H5, H6). 

When it comes to peer relationships, a review suggests that the level of perceived rejection by peers and the level of disliking of classmates may foster cyberbullying behaviors (H7, H8). Additionally, it seems that a perceived threat from peers will exacerbate the propensity to become a cyberbully (H9). Given the inconclusive findings on the role of peer group support in cyberbullying, a research question was posed as to whether cyberbullying is related to peer group support (Q3).

The study specifically addressed the importance of the elements of the immediate context of cyberbullying—that is, the selected aspects of problematic Internet use. Based on a review of previous research, it was assumed that both excessive use and irritability in the situation of Internet withdrawal would be positively correlated with the use of cyberbullying (H10, H11).

## 2. Materials and Methods

### 2.1. Participants, Procedure and Context of the Study

The research was conducted as part of the Ostroleka Educational Observatory project implemented by the Faculty of Pedagogy of the University of Warsaw in cooperation with the local government of Ostroleka (see: www.pedagog.uw.edu.pl/ostroleka/, accessed on 26 October 2022), which is a city in central-eastern Poland with a population of more than 50,000, with negative dynamics (5.2%, close to the average for small and medium-sized cities in Poland) between 2002–2021. In terms of the education of the city’s adult population (higher education 22%) and salary, the city ranks slightly above the national average (according to the ranking Poland in Numbers. https://www.polskawliczbach.pl/, accessed on 26 October 2022).

The town manages 28 educational institutions, including eight elementary schools (one special needs school). The study population consisted of eighth-grade students (aged 14–15) from seven elementary schools. The results of the end of primary school exams indicate that in the schools surveyed the distribution of students’ achievements in mathematics was very close to the nationwide distribution, and in Polish language it shifted slightly towards high scores (according to data and a report by the Central Examination Board in Warsaw).

The size of the target population was 698. The study group thus represents nearly 78% of the target population (N = 541), with a slight overrepresentation of girls. The study involved a local school population but given the demographic and educational characteristics of Ostroleka it can be assumed that the results obtained are typical for Poland. The characteristics of the respondents are shown in Table 1.

Data were collected in the spring of 2022 using computer-based online questionnaires completed by each student during school hours in the presence of a trained interviewer. The written consent of parents was obtained before the survey. Then students were asked for consent. The project was favorably reviewed by the Research Ethics Committee at the Faculty of Pedagogy of the University of Warsaw.

### 2.2. Measures

The tools used in the study are self-report scales. All but one (the Self-Regulation Scale) are from the Battery of Social Functioning Questionnaires developed for the diagmatic system for psychopedagogical diagnosis [120]. The content characteristics of the measurement scales applied are presented first, followed by the psychometric characteristics.

#### 2.2.1. Cyberbullying Perpetration and/or Victimization

The measurement of cyberbullying referred to Willard’s concept [7] and included: *harassment* (sending repeated, offensive messages), *exclusion* (excluding someone from a group of friends), *flaming* (an online fight), *outing and trickery* (revealing private information without someone’s permission), *cyber-stalking* (blackmailing someone by, for example, posting compromising photos online), *denigration* (posting false information about someone’s life), and *impersonation* (pretending to be someone in order to bring them into conflict with others).

The study used two scales of perpetrating and experiencing cyberbullying from the Disorders in Media-related Functioning Questionnaire [120]. Each consisted of seven items, to which the respondents gave answers using a five-point scale ranging from ‘never’, to ‘four times or more’. Indicators of violence taking place in the private sphere were included (experiencing cyberbullying: “How many times, in the past 6 months, have you received abusive text messages from someone for an extended period of time”) and public (use of cyberbullying: “How many times, in the past 6 months, have you posted disparaging photos of a person on the Internet without their permission (e.g., on social media”).

#### 2.2.2. Personal Resources

The Self-Regulation Scale (SRS) [121], specifically the Polish adaptation developed by Gajda [122], was used to measure emotional self-regulation. The abbreviated version of the emotional dimension of the SRS consists of 4 items (for example: “I have difficulty controlling my temper.”) to which interviewees respond using a 4-point scale ranging from 1 (Strongly agree) to 4 (Strongly disagree). For the purposes of the analysis, the results were recoded so that higher scores signified higher levels of self-regulation. The shortened version of the emotional dimension of the SRS demonstrates reliability and validity indices equivalent to the original longer (8 items) version compare Table 2 from [122].

The Individual Functioning Questionnaire [120] was used to measure self-esteem (e.g., “I am pleased with myself”), internal locus of control (e.g., ”The effect of my work depends mostly on how well I plan it”), and optimism (e.g., ”I will certainly achieve what I want.”). Each scale consists of 4 items to which answers are given on a 5-point scale ranging from 1 (I disagree) to 5 (I agree). A higher score indicates a higher level of the trait.

#### 2.2.3. Social Skills

Four social skills were measured by the Social Relationship Functioning Questionnaire [120]: prosocial behavior (4 items, for example: “I am happy to help if someone needs it”), assertiveness (5 items, for example: “I usually say directly what I think”), cognitive empathy (4 items, for example: “I easily recognize people’s feelings by looking at their faces”), and cooperation skills (4 items, for example: “I am able to reconcile people in my team who have quarreled”).

#### 2.2.4. Peer Relationships

Peer group functioning was shown from the perspective of resources (feeling supported by a peer group) and deficits (feeling threatened, rejected by peers, and having negative attitudes toward peers). The Peer Support scale is derived from The Educational Relationship Functioning Questionnaire [120] and consists of 6 items (e.g., “I can count on my peers in any situation”) with a 5-point response scale. Higher scores indicate higher levels of perceived support.

The Disrupted Peer Role Questionnaire [120] allowed for the measurement of feelings (real or imagined) of being under threat from peers (5 questions, e.g.,: “At school, I am often exposed to taunts from other students”), feelings of rejection from peers (4 questions, such as: “In many situations at school, no one wants to cooperate with me”) and negative attitudes toward peers, disliking them (5 questions, e.g.,: “I dislike many of my classmates”). Answers are given on a 5-point scale from 1 (I disagree) to 5 (I agree). Higher scores indicate higher levels of deficit.

#### 2.2.5. Problematic Internet Use (PIU)

Two PIU indicators were created on the basis of items from the Disorders in Media-related Functioning Questionnaire [120]:

Excessive Internet use, measured with 5 items with a 5-point response scale (e.g., “I often use a smartphone, tablet or computer instead of studying”), refers to the misuse of the Internet in terms of time, compulsive use, and loss of control over time spent online, with the result that an individual’s life becomes dominated by or subordinated to the Internet.

Impulsive reactions to Internet deprivation, measured with 4 items on a 5-point response scale (e.g., “I sometimes explode with anger when I can’t play on my smartphone, tablet or computer”), refers to the loss of control over emotions when it is impossible to use the Internet.

For both scales, higher scores indicate higher levels of PIU.

#### 2.2.6. The Factor Structure and Psychometric Properties of the Applied Scales

A confirmatory factor analysis (CFA) was performed to confirm the factor structure of the dependent variable and predictors. The CFA was conducted on a polychoric correlation matrix [123,124] using the weighted least squares with a mean- and variance-adjusted (WLSMV) estimator [125] to account for polytomous items [126].

For each construct, we tested the (assumed) single-factor model [120]. The CFA results were evaluated on three goodness-of-fit statistical measures: (1) root mean square error of approximation RMSEA—[127], (2) Tucker-Lewis index TLI—[128], and (3) comparative fit index CFI—[129]. According to the accepted rules [130,131], we assumed that a model that is adequately fitted to the data should be characterized by RMSEA values that are equal to or lower than 0.08, and values of CFI and TLI higher than 0.90. The modelling was performed using the R [132] package Lavaan [133].

Reliability was analyzed using an alpha internal consistence coefficient (based on the Pearson correlation matrix) and McDonald’s omega coefficient [134] (based on the polychoric correlation matrix). Simulation studies suggest that the alpha coefficient underestimates true reliability unless the items are tau equivalent, i.e., that the factor loadings are equal [135]. The latter coefficient does not require the assumption of essential tau-equivalence. Therefore, the omega coefficient is deemed to be a practical alternative to the alpha coefficient in estimating measurement reliability of the total score [136]. Alpha and omega coefficients over 0.70 indicate that the scale is reliable [137]. Both the alpha and omega coefficients were calculated using Lavaan [133] and the Bifactor Indices Calculator [138], operating in the R environment [132].

The reliability of all the measures used in the regression analyses proved at least satisfactory. Cronbach’s alpha for internal consistency coefficients ranged from 0.64 to 0.93 (*M* = 0.77). The omega reliability coefficients were also within a satisfactory (range 0.63 to 0.93; *M* = 0.77). In addition, the results of the confirmatory factor analyses showed a good fit of the data to the univariate models tested (although in some cases it was necessary to correlate the residuals). In the final models, the RMSEA did not exceed the value of 0.049, while the CFI and TLI did not fall below 0.970. Detailed data are shown in Table 2.

### 2.3. The Plan of the Main Analyses

Due to the right-skewed distribution of the dependent variable (see Figure 2), two-part regression modelling [139] was used in the analyses. This model allows for a better understanding of the results when, for example, a significant proportion of respondents do not show the presence of a particular phenomenon (relatively low prevalence), e.g., alcohol consumption, drug use, being a cyber-perpetrator, or bully victimization [140]. Generally, this model allows us to check whether the predictors have the same or different effects on: (1) the occurrence of a phenomenon (does not appear vs. appears) and (2) the frequency of its occurrence. In the case of interest to us, the use of the two-part regression allows us to check whether the likelihood of using cyberviolence and how often it is used may be affected by other factors.

The idea behind the model is to simultaneously test the predictors of two forms of the dependent variable: the dichotomous and the continuous. In the dichotomous part (logistic regression model), we are looking for the factors that influence the occurrence or non-occurrence of an outcome. In the continuous part, we are interested in predictors of the intensity of the phenomenon of interest if it occurs—that is, excluding cases where the phenomenon does not occur [141]. This corresponds to the hurdle model [142], except that the dependent variable does not have to be expressed as an integer—it can also be any rational number.

In the model we used, the dichotomous part refers to the prediction of being the perpetrator of cyberbullying regardless of its frequency and form (any use). The continuous part was used to examine the factors that have an effect on the frequency of a person becoming a perpetrator of cyberbullying, provided that the respondent declared that they had been a perpetrator of at least one of the seven forms of cyberbullying. As shown in Figure 2, nonzero values exhibit positive skewness. Therefore, in our analyses we decided on log transformation of the data (in the continuous part of the model), which is the default procedure used in Mplus [141].

Regression analyses were performed with Mplus 8.3 [143] using maximum likelihood estimation with robust standard errors (MLR), which is more than classical maximum likelihood estimation (ML), robust to violations of nonnormality and some forms of model misspecification, such as heteroscedasticity [141].

As is typical of studies conducted in the educational context, data in the present study were hierarchical—children were nested in classes. To correct for potential bias associated with class membership, we performed all analyses using the complex sample option in Mplus, which takes into account any clustering of the data and thus helps to avoid bias towards standard errors and test statistics [144].

The following Figure 1 shows a grid of explanatory variables (personal resources, social skills, peer relations, and problematic Internet use) and control variables (cyberbullying victimization, gender).

Four two-part regression models were used in the analyses of the determinants of cyberbullying perpetrator behaviors. The first control model included only gender and cyberbullying victimization in the pool of independent variables. This model, with its novel approach to the operationalization of being a perpetrator (splitting the variance of the explained variable into dichotomous and continuous parts), made it possible to confirm the well-documented, strong relationship between being a perpetrator and being a victim of cyberbullying. At the same time, this made it possible to test the relationship between gender and cyberbullying.

The second model included the following variables from the sets of independent variables: personal resources, social skills, peer relations, and control variables. Given an extensive set of variables, it made it possible either to answer the questions posed or to verify the hypotheses. This model did not yet include factors directly related to Internet use.

The third and fourth models introduced two variables directly related to Internet use. In the first step, excessive Internet use was included in the set of explanatory variables, while in the second step impulsive reactions to restricted Internet access were also included. These models were designed to test the extent to which the correlations detected in the second model could be mediated by the addictive use of the Internet.

## 3. Results

### 3.1. Frequency of Cyberbullying and Its Various Forms

Preliminary analyses included calculating means (*M*) and standard deviations (*SD*) or percentages, as appropriate, for all study variables, and Pearson product-moment correlations between the key constructs. These analyses were performed using the apaTables package [145] under the R environment [132].

An analysis of the students’ responses to the question about the frequency of their use of the various forms of cyberbullying showed quite significant variation (see Table 3). For all seven forms of cyberbullying analyzed, a one-sample *t*-test was calculated to check which of them was used significantly more often than others, i.e., which of them differed from the overall mean (the mean for all forms). Since for each form of cyberbullying the score could range from 0 to 4, the overall mean could also take values in this range. In fact, it was 0.35 (*SD* = 0.56).

The *t*-test for one sample revealed that two out of seven forms of cyberbullying, i.e., harassment (*M* = 0.78, *t*(546) = 7.32, *p* < 0.001, *d* = 0.31) and exclusion (*M* = 0.61, *t*(546) = 4.98, *p* < 0.001, *d* = 0.21) were mentioned more often than others. The severity of one form (flaming) did not differ from the overall mean (*M* = 0.41, *t*(546) = 1.44, *p* = 0.150, *d* = 0.06), whereas four of the forms were indicated relatively rarely: impersonation (*M* = 0.11, *t*(546) = −11.05, *p* < 0.001, *d* = −0.47), denigration (*M* = 0.14, *t*(546) = −8.89, *p* < 0.001, *d* = −0.38), cyberstalking (*M* = 0,14, *t*(546) = −8.94, *p* < 0.001, *d* = −0.38), and outing and trickery (*M* = 0.25, *t*(546) = −3.02, *p* = 0.003, *d* = −0.13).

From the point of view of the statistical analyses, it is significant that almost half of the students (49.2%) declared that they had not engaged in any cyber-related activities in the previous six months (Figure 2), as a result of which the distribution of the dependent variable deviated significantly from normal (*t*(547) = 0.291, *p* < 0.01).

### 3.2. Descriptive Statistics and Correlation Matrix

The descriptive statistics and Pearson’s r correlation coefficients for all variables used in the analyses are presented in Table 4. Due to the skewness of the distributions of the variables describing cyberbullying, the calculated values of correlation coefficients for them should be treated as approximations.

### 3.3. Predictors of Cyberbullying

The two control variables (being a victim of cyberbullying, and gender) included in Model 1 (Table 5 and Figure 3) significantly modified the level of the dependent variable in both the binary and continuous parts of the model. Being a victim and a male increased the chances of not only being involved in at least a single episode related to being a perpetrator of cyberviolence (the dichotomous part of the model), but also increased the number of such episodes (the intensity of being a perpetrator; the continuous part of the model). At the same time, a comparison of the values of the regression coefficients indicates that, in both parts of the model, being a victim was a stronger predictor than gender. Both control variables explain a significant portion of the variance in the dependent variable: 24% in the dichotomous part of the model and 32% in its continuous part.

The inclusion of variables related to personal resources, social skills, and peer relationships in the next model (Model 2; see Table 5 and Figure 4) significantly increased the percentage of explained variance in both the dichotomous and continuous parts of the model (up to 42% in both). Of the four personal variables—when controlling for the other variables entered into the model—only emotional self-regulation was found to be significantly associated with being an offender. Negative beta coefficients indicated that higher levels of emotional self-regulation reduce both the probability of even a single episode related to being a perpetrator of cyberviolence (the dichotomous part of the model) and the number of such episodes (the intensity of being a perpetrator; the continuous part of the model).

In contrast, of the four social skills variables, two-prosocial behavior and assertiveness significantly translated into being a perpetrator of cyberviolence. Interestingly, while a higher level of prosociality lowered the chances of being involved even in a single episode of cyberviolence (the dichotomous part of the model), higher assertiveness increased the likelihood. None of the variables describing students’ social skills translated into the number of episodes associated with being a perpetrator of cyberviolence (the continuous part of the model).

A similar pattern of correlations occurred for variables related to peer relations. Again, only two of them, peer support and dislike of peers, were found to be significantly associated with being a perpetrator. At the same time, while disliking one’s peers was associated with being a cyber-perpetrator only in the dichotomous part of the model, peer support increased not only the chances of being a cyber-perpetrator but also the frequency of such behavior.

It is still worth noting that while being a cyber-victim remained significantly associated with being a perpetrator in both the dichotomous and continuous parts of the model, once the new variables were introduced, gender ceased to be a significant predictor in the dichotomous part while still significantly interacting in the continuous part. When controlling for the variables describing personal resources, i.e., social skills and peer relationships, male students were more often involved in cyberbullying acts.

In the next model (Model 3; Table 5 and Figure 5), the variable of excessive Internet use was added. It proved to be a significant predictor of a person being a perpetrator of cyberviolence, although only for the dichotomous part of the model. In the continuous part, a confidence interval “hooked” to zero dictates caution. Overall, however, it can be assumed that excessive Internet use translated into a higher probability of cyberviolence.

Notably, adding a new predictor to the model hardly changed the impacts of other variables. When controlling for excessive Internet use, prosocial behavior still reduced the chances of involvement in cyberviolence, whereas assertiveness, peer support, and disliking one’s peers increased these chances. In the dichotomous part of the model, only emotional self-regulation (which in the previous model reduced the likelihood of cyberviolence episodes) ceased being significantly related to being a cyber-perpetrator.

The inclusion of a new variable (excessive Internet use) in the model also resulted in relatively small changes in the continuous part of it. Being a victim of cyberviolence, being a male, and emotional self-regulation were still significantly associated with the frequency of being a perpetrator. The first two are risk factors, while emotional self-regulation is a protective element. Compared to Model 2, only peer support lost statistical significance. Additionally, the variables included in the model explained 45% of the variance of the dependent variable in the dichotomous part and 43% in the continuous part.

Another Internet-related variable introduced to the final model (Model 4; Table 5 and Figure 6) was ‘impulsive reactions to Internet deprivation’. As expected, the more impulsive the reactions to Internet deprivation, the greater the likelihood of even a single episode related to being a perpetrator of cyberviolence (the dichotomous part of the model). At the same time, impulsive response to Internet deprivation was not significantly linked to the number of episodes of being a perpetrator of cyberviolence (the continuous part of the model). Thus, Internet deprivation translated into being a perpetrator, but not into the number of episodes associated with being a perpetrator of cyberviolence. Significantly, with the inclusion of a new predictor in the model, the variable of excessive Internet use was no longer significantly associated with being a perpetrator of cyberviolence (dichotomous part).

The analysis of the results shows the importance of one more change in relation to the previous model. In the continuous part, we found that feeling threatened by peers reduced the involvement in cyberbullying acts in a statistically significant way. This is despite controlling for being a victim of cyberbullying (if not, the effect would be obvious: victims fear peers).

Summing up, in the results of the last regression model test that involved a broad set of independent variables, six variables turned out to be the risk factors for cyberbullying (the dichotomous part), i.e., being a victim of cyberbullying, assertiveness, peer support, disliking peers, and impulsive reactions to Internet deprivation. Only one predictor—pro-social behavior—proved to be a protective factor.

In contrast, four factors translated into the frequency of being a perpetrator of cyberviolence (continuous part). Apart from being a victim and a male, which increased the number of episodes of cyberviolence perpetration, these were: emotional self-regulation and feeling threatened by peers, which reduce the number of such episodes. The predictors in Model 4 explained 47% of the variance in the dependent variable in the dichotomous part and 44% in the continuous part.

## 4. Discussion

In the surveyed population of a medium-sized city in Poland, almost half of the students in the eighth grade (49.2%) declared that they had not engaged in any cyberbullying as a perpetrator in the previous six months, i.e., slightly more than half were involved in at least one such episode during that period. An analysis of the students’ responses to the question about the frequency of their use of various forms of cyberbullying shows quite considerable variation. Two of the seven forms of cyberbullying, i.e., harassment and exclusion, were mentioned more frequently than others. The level of intensity of flaming was not significantly different from the overall average, while four forms were indicated relatively infrequently, i.e., impersonation, denigration, cyberstalking, and outing and trickery. Our findings with regards to the frequency of various forms of cyberbullying are typical of the Polish adolescent population [118,146]. Notably, the types of online violence included in the study are arranged on a continuum of private-to-public space, and their placement is correlated with their declared frequency by the respondents.

The model testing the significance of the control variables yielded two important results. First, it confirmed the strong association between being a perpetrator of cyberbullying and having a prior experience of being a victim. The effects for both zero-one cyberbullying (the binary part of the model) and its frequency (the continuous part of the model) are significant. This relationship also holds when a wide range of student characteristics are included in the analysis models. This is another strong support for the Cyclical Violence Model [18]. Second, the gender variable proved to be a significant predictor of cyberbullying. However, the model including the full set of explanatory variables showed that the relationship held only for the frequency of cyberbullying acts with a higher occurrence to be expected in boys. Engaging in cyberbullying itself vs staying away from it was not linked to gender.

Within the group of constructs describing students’ personal resources, our analyses showed that, when controlling for the variables of being a victim and gender as well as a wide range of factors describing social skills and peer relationships, the only significant personal resource for engaging in cyberbullying is emotional self-regulation (H1). The three other variables, including self-esteem, are not significant predictors of being a perpetrator of cyberbullying (H2, Q1, Q2). That is, the prediction of a negative association between self-esteem and the perpetration of cyberbullying has not been confirmed. An analysis model that additionally considers the factors describing the immediate context of cyberbullying—excessive Internet use and impulsive response to constraints (deprivation)—has confirmed the direct, protective effect of self-regulation, but indicated that it is more significant for the frequency of cyberbullying acts than for zero-one-defined engagement in cyberbullying (the latter effect was not statistically significant). This is despite the presence in the model of a variable describing an emotional indicator of Internet addiction-impulsive reactions to Internet deprivation.

The protective effect of self-control confirmed in the study is consistent with the theoretical, even definitional prediction of its importance in inhibiting cyberbullying. If the Cyclical Model of Violence [29] is accurate, such a result was to be expected. However, doubts were already signaled in the introduction. Both the theoretical rationale (the interaction of narcissism with self-regulation) and research findings suggest that high self-esteem may be both conducive or unrelated to cyberbullying [18,28]. Additionally, a recent longitudinal study [147] found that while compulsive Internet use preceded the development of some aspects of emotion dysregulation, there was no evidence that difficulties in emotional regulation were the cause of addiction. Admittedly, Internet addiction is not the same as cyberbullying, but a certain affinity of mechanisms would be expected. However, the result obtained in this study should be taken as an argument for the thesis that cyberbullying has a significant emotional component [148].

Another result worth commenting on is the lack of a significant relationship between self-esteem and perpetration of cyberbullying, which is not entirely surprising. Although the introduction tended to favor the thesis that high self-esteem is a protective factor, it is worth recalling that a meta-analysis of longitudinal studies indicated a lack of correlation [35], i.e., the result obtained in the study is not a rarity, but confirms the lack of a direct, straightforward relationship between self-esteem and cyberbullying.

Let us move on to social skills. The results of the study allow us to accept only some of the hypotheses regarding social skills as predictors of cyberbullying behavior (H3–H6). It turns out that of the four variables (empathy, assertiveness, pro-social behavior, and cooperativeness), only two, i.e., assertiveness and pro-social behavior, are associated with cyberbullying in each of the multivariate regression models tested (in the binary part). High assertiveness is among the risk factors for such behavior, while prosociality turns out to be a protective factor against cyberbullying. It should be noted, however, that our study suggests that none of the variables describing students’ social skills translate into the frequency of cyberbullying acts (the continuous part of the models).

The results obtained in our study regarding the association of two social skills with cyberbullying are not surprising: both the effect for assertiveness [2,8] and prosociality are consistent with the previous literature [34,62,68,69,149,150]. In contrast, we find that the lack of an association of cognitive empathy and cooperativeness with cyberbullying found in this study is not clearly supported by previous research.

Previous findings indicate that cooperative skills and friendly interpersonal relationships among adolescents are effective in reducing cyberbullying behavior [34,62,151]. Hence, the lack of a statistically significant relationship between cooperativeness and cyberbullying yielded by the present study comes as a certain surprise and suggests the need for further research.

Next comes empathy, an intuitive ability that has been associated with inhibiting violence. Some of the literature considers low levels of affective and cognitive empathy to be predictors of more frequent involvement in cyberbullying [34,152,153,154], but the results of other studies suggest that perpetrators of cyberbullying exhibit higher levels of these traits [2]. In contrast, the lack of a significant relationship found in the present study is consistent with the findings by Kowalski, Giumetti, Schroeder, and Lattanner [8] and Zych, Farrington, and Ttofi [34]. 

In fact, in order to explain the lack of a relationship between empathy and cyberbullying, reference can be made to the studies by Lockwood, Hamonet, Zhang, Ratnavel, Salmony, Husain, and Apps [155]. They have demonstrated that when affective empathy is too high, an individual ignores other people’s suffering to cope with their own emotions. In contrast, as Kool, McGuire, Rosen, and Botvinick [156] note, engaging in cognitive empathy is taxing on the brain, and studies show that individuals avoid cognitive tasks that require high effort. Simply put, cognitive empathy is exhausting, and individuals decide whether to invest in it before they engage in prosocial behavior [157]. Research clearly indicates that the greater the number of individuals suffering (e.g., as a result of cyberbullying), the less distress people feel and the less they engage in pro-social behavior [54], which our research has shown to be a statistically significant protective factor against cyberbullying.

In the case of peer relationships, higher levels of peer support increase the chances of being a perpetrator of cyberbullying (Q3). This relationship proved statistically significant in each of the regression models in the dichotomous part. In the continuous part, it disappeared when the variables related to the problematic use of the Internet were included (in the last two models). Therefore, our analyses confirm the findings of some research [158] to the effect that peer support can, by providing sense of security, push adolescents towards risky behavior, including cyberbullying.

This suggestion is in line with some studies, which indicate that bullies are popular among peers [159,160] and that peer groups perceived as highly popular tend to do a lot of bullying [78]. Therefore, bullying may be explained by the high status it confers on its perpetrators, which is something they aim for [161]. It might be speculated that the higher social status of perpetrators associated with peer support and the growth of power that perpetrators exert over their victims constitutes a prerequisite for bullying [162].

Notably, popularity is an indicator of admiration, reputation, social dominance, and power but it is not synonymous with being liked. While liking is a dyadic construct, reflecting an individual’s personal relationship with one of their peers, popularity is a belief shared within a group [163]. Previous research shows that although perpetrators are perceived (and perceive themselves) as popular, they are less liked and are expected to be more disliked by their schoolmates than other children [161,164,165], and that bullying is positively associated with a tendency to value being popular more than being liked [166]. These findings support the hypothesis that bulling is a way to achieve/maintain a dominant social position, i.e., social status (popularity) associated with greater social power [167,168,169,170,171].

At the same time current research, contrary to our hypothesis (H7), indicates that feelings of peer rejection are not associated with being a perpetrator (nor with cyberbullying as such). One possible hypothesis explaining why perpetrators might not be viewing their relationships with others as negative is that they hardly see themselves as rejected by their peers (despite potentially being liked less than others) and that the social status they have achieved is more important to them than their position in the affective network [166]. In this case, popularity can compensate for the deficit in likeability.

Moreover, our research shows that the perpetrators’ social support and not necessarily negative perception of their own peer relationships are linked to evaluating other students in their school more negatively. In line with our hypothesis, the research we have conducted confirms that for cyberbullying (but only in the dichotomous parts of each of the estimated models), disliking one’s peers is a risk factor (H8). What matters, then, is not so much how the offender perceives his position in the group (the cognitive aspect), but his dislike of peers (the affective aspect), which may be a reaction to peer rejection [163,172]. Disliking a person with whom one has to interact on a daily basis, such as a classmate, is psychologically stressful [173], while aggressive behavior, especially at a heightened level of emotional instability [174], may be a maladaptive coping style [175]. The fact remains that it is resentment toward peers that proves to be one of the factors activating violent behavior.

Interestingly, the relationship between feeling threatened by peers and being a perpetrator of cyberbullying took the opposite direction to that assumed in this study (H9). When controlling for being a victim, gender, personal characteristics, interpersonal skills, perceptions of peer relationships, and Internet use, higher levels of perceived threat from peers reduce the extent and/or frequency of forms of cyberbullying used (although it does not translate into the likelihood of being or not being a perpetrator of cyberbullying). It may be that previous victimization causes retaliation, but with a relatively low level of activity.

Finally, our attention shifts to the variables introduced into the explanatory model that describe the immediate context of cyberbullying, i.e., problematic Internet use (H10, H11). Our analyses only partially confirmed the hypotheses regarding the use of the Internet. An excessive engagement with the Internet increased the likelihood of cyberbullying but was not significantly associated with the frequency of such behavior (the severity of the problem). In addition, the relationship was no longer significant when impulsive responses to Internet deprivation were included in the explanatory model. Emotional control disorders associated with the inability to use the Internet proved to be significantly associated with the likelihood of finding oneself in a cyberbullying space, but also failed to predict the number of episodes associated with being a perpetrator. Thus, of the two variables describing problematic Internet use, only impulsive reactions to Internet deprivation proved to be associated with committing online violence, and none of the variables analyzed translated to the number of episodes of cyberbullying.

In the introductory section, it was noted that previous correlational studies have indicated a weak positive association between problematic Internet use and being a perpetrator of cyberbullying [5,8]. However, they differed from ours in that they failed to include many important control variables, including being a victim of cyberbullying [108,110,176]. The likelihood of experiencing cyberbullying is increased by the risk of problematic Internet use [177], so it is probable that previous correlational studies that did not control for being a victim of cyberbullying overestimated the importance of this group of factors for becoming a perpetrator.

Our study has shown that excessive use of the Internet while controlling for the emotional reaction associated with Internet deprivation is not associated with the perpetration of cyberbullying, which is consistent with reports from longitudinal studies to the effect that the amount of time spent online is irrelevant as far as the development of cyberaggression is concerned [106,107]. On the other hand, it has been shown that impulsive reactions to Internet deprivation promote the perpetration of cyberbullying. Note that impaired emotional control in situations of Internet deprivation can be regarded as a manifestation of impulsivity or lack of self-control, but they are also responses typical of an addict [98]. In our study, we controlled for overall levels of emotional self-control, so the latter track seems reasonable. Some studies have shown that adolescents addicted to the Internet are more likely to exhibit aggressive behavior [178]. However, others have found that common neurobiological characteristics and mechanisms underlie Internet addiction and aggressive behavior [115,116]. The inclusion of two indicators of problematic Internet use in the study thus allowed us to see that the problem may be more complex. It may be particularly important to distinguish between excessive Internet use and Internet addiction. Previous studies have not been able to show these differences, as they included one general indicator of this syndrome [108,110,176]. Considering the complexity of the phenomenon and the prior level of aggressive behavior could be an important direction for further research on the determinants of cyberbullying associated with problematic use of digital technologies. However, it should be emphasized that the effect obtained in the study had moderate strength and was reflected only in the dichotomous part of the model.

The results presented also have practical significance, suggesting the areas in which psychologists, educators, teachers, and parents may become more aware and actively involved in the prevention of cyberbullying. Because our research indicates that (1) being a victim of cyberbullying is the most important predictor of being a perpetrator (cyberbullying may therefore be a maladaptive response to problematic situations) and (2) being a perpetrator is associated with the variables relating to emotional regulation in the broadest sense (such as emotional self-control, assertiveness, resentment toward peers, or impulsive responses to Internet use restriction), effective interventions aimed at reducing the intensity of cyber-perpetration should therefore focus on the development of more adaptive styles of anger expression [179,180]. In this context, a review of the available meta-analyses by Lee and DiGiuseppe [181] indicates that “cognitive-behavioral and other approaches to treatment of anger are comparable and at least moderately effective, with limited evidence favoring relaxation treatments over other types” [181]. The training offered on the adaptive styles of responding to problematic situations should not be limited to offline violence, but it should also include violence and emotions occurring in the online environment [26]. It is worth making parents and teachers aware that reducing teenager’s time online is not an effective remedy against cyberbullying.

Finally, we come to the limitations of this study. First, the study population was eighth-grade students in one medium-sized city, so its findings cannot be generalized to all teenagers in Poland, even though the demographic and educational characteristics of the city suggest that the study group was representative.

For testing causal hypotheses, an obvious limitation is its cross-sectional nature, so any conclusions about causal relationships between the variables should be treated with caution (hence the authors’ caution when it comes to using the language of causality). However, it should be also remembered that 14 independent variables and two control factors were included in the model describing the determinants of cyberbullying, which certainly increases our confidence in the relationships discovered.

Although the data analyzed meet the standards of psychometric quality, its shortcoming is that it is the result of using self-reporting tools. For constructs relating to the feelings and opinions of respondents, this way of collecting data is natural, but when we are interested in the behavior of individuals it poses a serious threat to data accuracy. However, observational methods in the study of cyberbullying are extremely difficult and using them encounters barriers of time and high cost, which would radically limit the sample size and the statistical power of empirical generalizations.

In terms of the potential factors responsible for cyberbullying that are absent from the study discussed here, one must point out the lack of control of traditional aggression and potential protective/threatening factors related to family and peer environments. The family should certainly address methods of controlling an adolescent’s online behavior, while the clues emerging in this study regarding peer relationships point to the need to address group processes, e.g., by measuring sociometric popularity. A final factor absent from this study—which the authors believe is worth pursuing in future research—is including the various dimensions of cyberbullying in its measurement, e.g., reactive vs. proactive aggression, instrumental vs. hostile aggression, analogously to research on offline aggression.

## 5. Conclusions

The survey found that in the population of students in the eighth grade (modal age 15) of a medium-sized city in Poland, the prevalence of cyberbullying is significant. Half of the teenagers declared that they had been involved in at least one episode of cyberbullying as a perpetrator in a six-month period. The most common form of cyberbullying was harassment and exclusion, the least common were cyberstalking, denigration, and impersonation. The study confirmed that being a perpetrator is strongly correlated with an experience of being a victim, and that the correlation is strongest across a broad set of psychosocial characteristics of adolescents. This confirms the validity of including cybervictimization as a control variable in the analysis model.

The distribution of the multi-item cyberbullying index clearly showed the non-continuity of the phenomenon. To adjust the analysis model for the fact that the study group broke into two subgroups—those who shunned cyberbullying and those who engaged in it, albeit with varying frequency—a two-part regression model was used. It allowed for avoiding the need to omit some of the variance in the cyberbullying indicator inevitably associated with the categorization of the dependent variable. The results indicate the effectiveness of this approach. Some factors proved significant in predicting that a teenager will remain outside the cyberbullying space, while others predicted the frequency of violent acts once a teenager was already in that space.

Our research, thanks to the simultaneous inclusion of a very wide range of characteristics of adolescents, whose association with cyberbullying had been studied before, provided an opportunity to detect the factors crucial to cyberbullying. Controlling for so many variables in the model, including those directly linked to the context of Internet use, made it possible to detect aspects that are directly, and not through the mediation of other variables or the mechanism of apparent correlation, likely to affect cyberbullying.

In light of our findings, the hypothesis that an emotional component is crucial to cyberbullying has gained even more support. This is indicated by the importance of emotional self-control (frequency of cyberbullying), assertiveness, impulsive response to limited Internet access (engaging in cyberbullying), and fear of peers (frequency). In turn, the importance of pro-sociality (which inhibits engagement) and peer support (which promotes engagement) points to a second important component of cyberbullying—namely, group mechanisms. This is a clue that requires further research to clarify the social factors responsible for this problem.

The introduction of variables related to problematic Internet use into the explanatory model made it possible to show that, although the importance of Internet addiction as a risk factor for cyberbullying should not be underestimated, the time spent online cannot be seen as the main source of the problem.

The results obtained are of methodological significance due to the use of two-part regression for the statistical description of cyberbullying. Of theoretical importance is the confirmation of the relevance of the emotional control factor and its link to group mechanisms. Finally, its practical relevance lies in pointing to emotional self-control training as a promising form of reducing the severity of cyberbullying.

## Figures and Tables

**Figure 1 ijerph-20-05521-f001:**
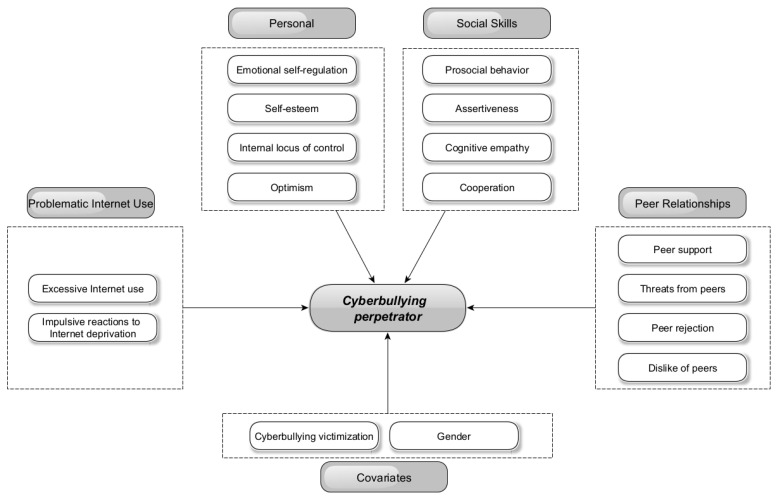
The grid of explanatory and control variables used in the cyberbullying analyses.

**Figure 2 ijerph-20-05521-f002:**
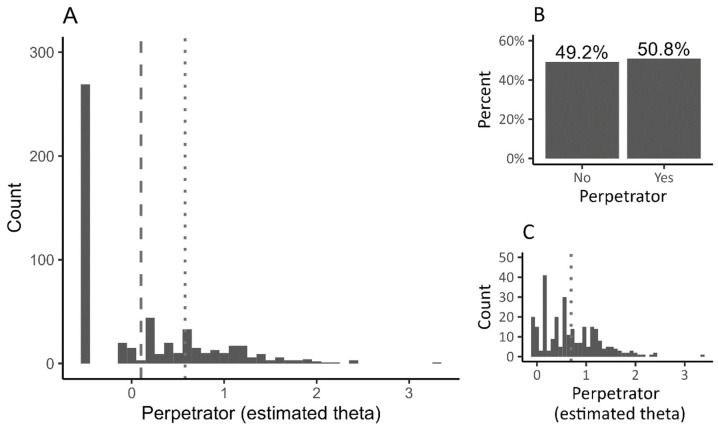
Distribution of the respondents according to being a perpetrator. Note: Panel (**A**) shows the distribution of the latent variable that illustrates the intensity of being a perpetrator (all respondents). Panel (**B**) shows the percentage of respondents who selected ‘Never’ for each of the six types of cyberbullying (category: ‘No’) and those who had engaged in some form of cyberbullying (category: ‘Yes’). Panel (**C**) illustrates the distribution of the latent variable of respondents who had participated in some form of cyberbullying (that is, after removing respondents who had never participated in any form of cyberbullying). The “dashed” line indicates the mean for all individuals (*M* = 0.10). The “dotted” line indicates the mean after eliminating respondents not involved in being the perpetrator (*M* = 0.69).

**Figure 3 ijerph-20-05521-f003:**
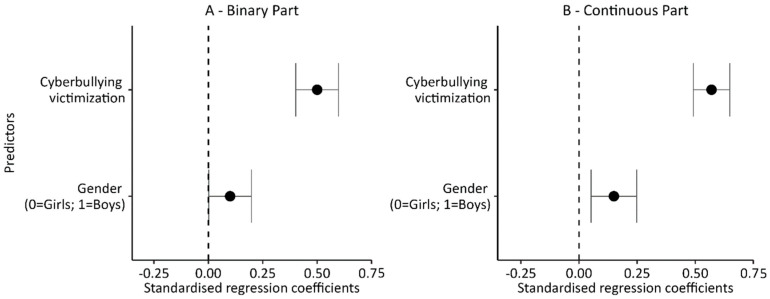
Model 1. Standardised regression coefficients with 95% confidence intervals. Note: Black circle-coefficient significant at 0.05.

**Figure 4 ijerph-20-05521-f004:**
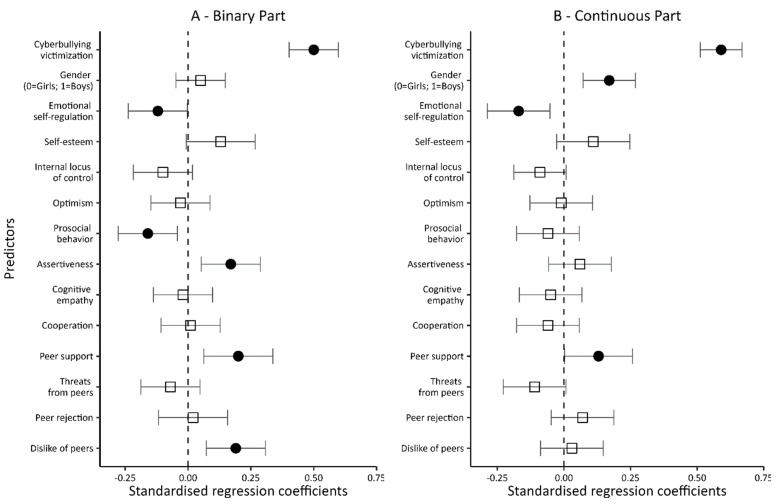
Model 2. Standardised regression coefficients with 95% confidence intervals. Note: black circle-coefficient significant at 0.05; white square-coefficient insignificant (*p* > 0.05); tails = standard errors.

**Figure 5 ijerph-20-05521-f005:**
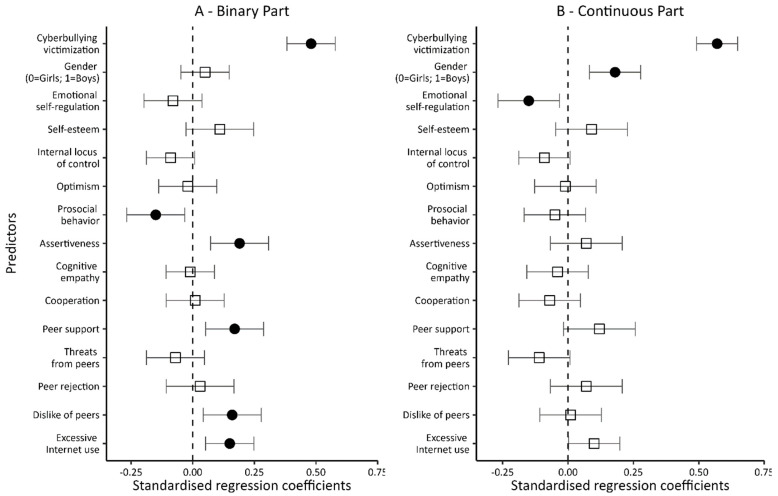
Model 3. Standardised regression coefficients with 95% confidence intervals. Note: Black circle-coefficient significant at 0.05; white square-coefficient insignificant (*p* > 0.05); tails = standard errors.

**Figure 6 ijerph-20-05521-f006:**
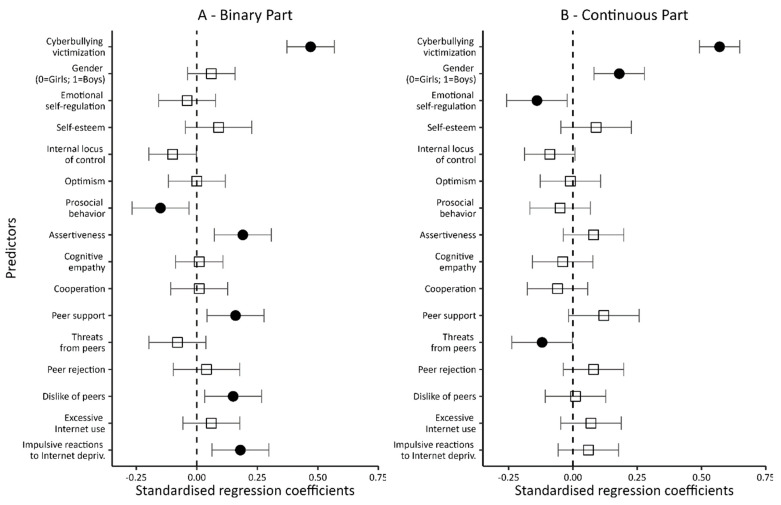
Model 4. Standardised regression coefficients with 95% confidence intervals. Note: Black circle-coefficient significant at 0.05; white square-coefficient insignificant (*p* > 0.05); tails = standard errors.

**Table 1 ijerph-20-05521-t001:** Population and characteristics of the respondents.

Characteristics of the Study Population	Numbers	Percentage
Size of the entire population	698	100.0
Female students in population	346	49.6
Students not included in the survey due to lack of parental consent or absence from school	133	19.1
Data excluded due to unreliability	24	3.4
Size of the study group	541	77.5
Female students in the study group	284	52.5
Students aged 14 in the study group	155	28.7
Students aged 15 in the study group	386	71.3

**Table 2 ijerph-20-05521-t002:** Reliability and model fit measures.

Variable	No. of Items	Cronbach Alpha	Omega	RMSEA	CFI	TLI
Cyber-Perpetrator	7	0.71	0.75	0.049	0.980	0.970
Cyber-Victim	7	0.76	0.77	0.016	0.998	0.998
Emotional self-regulation	4	0.84	0.85	0.000	1	1
Self-esteem	4	0.93	0.93	0.000	1	1
Internal locus of control	4	0.64	0.65	0.045	0.996	0.988
Optimism	4	0.84	0.84	0.041	0.999	0.998
Prosocial behavior	4	0.73	0.74	0.000	1	1
Assertiveness ^A^	5	0.80	0.83	0.000	1	1
Cognitive empathy	4	0.72	0.74	0.035	0.999	0.996
Cooperation	4	0.67	0.70	0.014	1	0.999
Peer support ^A^	6	0.82	0.82	0.044	0.996	0.993
Threats from peers ^A^	5	0.85	0.83	0.045	0.998	0.996
Peer rejection ^A^	4	0.78	0.76	0.049	0.999	0.995
Dislike of peers ^B^	5	0.71	0.63	0.000	1	1
Excessive Internet use ^A^	5	0.74	0.70	0.000	1	1
Impulsive reactions to Internet deprivation	4	0.83	0.85	0.027	1	0.999

Note: ^A^ = Model fit after correlating the residuals of one pair of variables; ^B^ = Model fit after correlating the residuals of the two pairs of variables.

**Table 3 ijerph-20-05521-t003:** Distribution of the respondents according to use of forms of cyberbullying (perpetrator of cyberbullying) (N = 547).

Cyberbullying Forms:	*M*	*SD*	Percent				
			Never	One	Two	Three	Four or More
Harassment	0.78	1.36	68.6	11.0	6.2	2.9	11.3
Exclusion	0.61	1.19	72.8	12.1	4.9	2.4	7.9
Flaming	0.41	1.01	80.8	8.6	4.2	1.3	5.1
Outing and trickery	0.25	0.79	88.3	4.9	2.9	1.3	2.6
Cyberstalking	0.14	0.55	91.6	5.5	0.9	1.3	0.7
Denigration	0.14	0.57	92.9	3.7	1.5	1.1	0.9
Impersonation	0.11	0.51	93.8	3.5	1.3	0.7	0.7

Note: *M* = Mean; *SD* = Standard Deviation.

**Table 4 ijerph-20-05521-t004:** Means, standard deviations, and correlations.

Variable	*M*	*SD*	1	2	3	4	5	6	7	8	9	10	11	12	13	14	15	16
1. Cyber-Perpetrator	0.10	0.73																
2. Cyber-Victim	0.07	0.80	0.57 **															
3. Gender (0-Girls; 1-Boys)	0.50	0.50	0.01	−0.17 **														
4. Emotional self-regulation	−0.01	0.89	−0.27 **	−0.37 **	0.36 **													
5. Self-esteem	−0.03	0.87	0.04	−0.25 **	0.35 **	0.38 **												
6. Internal locus of control	−0.01	0.83	−0.17 **	−0.16 **	0.02	0.20 **	0.26 **											
7. Optimism	−0.01	0.90	−0.03	−0.23 **	0.18 **	0.30 **	0.69 **	0.36 **										
8. Prosocial behavior	−0.01	0.86	−0.20 **	−0.04	−0.19 **	0.08 *	−0.00	0.23 **	0.06									
9. Assertiveness	−0.01	0.87	0.21 **	0.00	0.23 **	0.08 *	0.41 **	0.16 **	0.42 **	0.02								
10. Cognitive empathy	−0.02	0.84	−0.00	0.08	−0.17 **	−0.07	0.04	0.21 **	0.16 **	0.43 **	0.19 **							
11. Cooperation	−0.00	0.85	−0.08 *	−0.08	−0.05	0.16 **	0.24 **	0.22 **	0.34 **	0.51 **	0.28 **	0.47 **						
12. Peer support	−0.00	0.91	−0.07	−0.27 **	0.02	0.21 **	0.37 **	0.26 **	0.37 **	0.30 **	0.23 **	0.24 **	0.37 **					
13. Threats from peers	0.03	0.87	0.12 **	0.38 **	−0.05	−0.29 **	−0.34 **	−0.13 **	−0.27 **	−0.12 **	−0.22 **	−0.10 *	−0.21 **	−0.44 **				
14. Peer rejection	0.02	0.85	0.04	0.25 **	−0.09 *	−0.27 **	−0.39 **	−0.08	−0.33 **	−0.15 **	−0.32 **	−0.15 **	−0.29 **	−0.56 **	0.65 **			
15. Dislike of peers	−0.00	0.87	0.25 **	0.27 **	−0.13 **	−0.40 **	−0.23 **	−0.24 **	−0.22 **	−0.33 **	−0.03	−0.05	−0.29 **	−0.51 **	0.41 **	0.44 **		
16. Excessive Internet use	0.03	0.87	0.33 **	0.25 **	−0.13 **	−0.43 **	−0.06	−0.11 **	−0.13 **	−0.13 **	−0.07	−0.08	−0.14 **	−0.05	0.12 **	0.06	0.23 **	
17. Impuls. react. to Int. depr.	−0.00	0.88	0.30 **	0.25 **	−0.11 *	−0.34 **	−0.07	−0.17 **	−0.14 **	−0.15 **	−0.08	−0.10 *	−0.15 **	−0.07	0.10 *	0.06	0.24 **	0.62 **

Note: *M* and *SD* are used to represent mean and standard deviation, respectively. * indicates *p* < 0.05. ** indicates *p* < 0.01.

**Table 5 ijerph-20-05521-t005:** Predictors of cyberbullying. Two-part regression models.

	Model 1		Model 2		Model 3		Model 4	
	Part		Part		Part		Part	
	Binary	Continuous	Binary	Continuous	Binary	Continuous	Binary	Continuous
Independent Variables	β (SE)	β (SE)	β (SE)	β (SE)	β (SE)	β (SE)	β (SE)	β (SE)
*Control variables*								
Cyberbullying victimization	0.50 (0.05) **	0.57 (0.04) **	0.50 (0.05) **	0.59 (0.04) **	0.48 (0.05) **	0.57 (0.04) **	0.47 (0.05) **	0.57 (0.04) **
Gender	0.10 (0.05) **	0.15 (0.05) **	0.05 (0.05)	0.17 (0.05) **	0.05 (0.05)	0.18 (0.05) **	0.06 (0.05)	0.18 (0.05) **
*Personal resources*								
Emotional self-regulation	–	–	−0.12 (0.06) **	−0.17 (0.06) **	−0.08 (0.06)	−0.15 (0.06) **	−0.04 (0.06)	−0.14 (0.06) **
Self-esteem	–	–	0.13 (0.07)	0.11 (0.07)	0.11 (0.07)	0.09 (0.07)	0.09 (0.07)	0.09 (0.07)
Internal locus of control	–	–	−0.10 (0.06)	−0.09 (0.05)	−0.09 (0.05)	−0.09 (0.05)	−0.10 (0.05)	−0.09 (0.05)
Optimism	–	–	−0.03 (0.06)	−0.01 (0.06)	−0.02 (0.06)	−0.01 (0.06)	0.00 (0.06)	−0.01 (0.06)
*Social skills*								
Prosocial behaviour	–	–	−0.16 (0.06) **	−0.06 (0.06)	−0.15 (0.06) **	−0.05 (0.06)	−0.15 (0.06) **	−0.05 (0.06)
Assertiveness	–	–	0.17 (0.06) **	0.06 (0.06)	0.19 (0.06) **	0.07 (0.07)	0.19 (0.06) **	0.08 (0.06)
Cognitive empathy	–	–	−0.02 (0.06)	−0.05 (0.06)	−0.01 (0.05)	−0.04 (0.06)	0.01 (0.05)	−0.04 (0.06)
Cooperation	–	–	0.01 (0.06)	−0.06 (0.06)	0.01 (0.06)	−0.07 (0.06)	0.01 (0.06)	−0.06 (0.06)
*Peer relations*								
Peer support	–	–	0.20 (0.07) **	0.13 (0.07) **	0.17 (0.06) **	0.12 (0.07)	0.16 (0.06) **	0.12 (0.07)
Threats from peers	–	–	−0.07 (0.06)	−0.11 (0.06)	−0.07 (0.06)	−0.11 (0.06)	−0.08 (0.06)	−0.12 (0.06) **
Peer rejection	–	–	0.02 (0.07)	0.07 (0.06)	0.03 (0.07)	0.07 (0.07)	0.04 (0.07)	0.08 (0.06)
Dislike of peers	–	–	0.19 (0.06) **	0.03 (0.06)	0.16 (0.06) **	0.01 (0.06)	0.15 (0.06) **	0.01 (0.06)
*Problematic Internet use*								
Excessive Internet use	–	–	–	–	0.15 (0.05) **	0.10 (0.05)	0.06 (0.06)	0.07 (0.06)
Impulsive reactions to Int. depr.	–	–	–	–	–	–	0.18 (0.06) **	0.06 (0.06)
R−Square	0.24	0.32	0.42	0.42	0.45	0.43	0.47	0.44

Note: ** Statistical significance level (*p* < 0.01).

## Data Availability

The data presented in this study are available on request from the corresponding author.

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
