# Peer review of "Psycho-Social Correlates of Cyberbullying among Polish Adolescents"

_ijerph, 2023, doi:10.3390/ijerph20085521_

Round 1

Reviewer 1 Report

The article is interesting and current. The subject is being treated correctly and deepening the importance of relating social skills to cyberbullying.

The scores of the CFI and TLI measures are quite high, which gives more value to the research. Also, they are properly cited. All the text is well cited, but for future publications I recommend that the references be more current.

Current issues are being dealt with and there are references to 1991, which makes no sense. It is not something negative, but a recommendation for the authors to try to update the bibliographical references as much as possible, since there are much more current authors who deal with this topic.

1. What is the main question that the research addresses?  What components related to cyberbullying are the most important and frequent when it comes to this problem  

2. Do you consider the topic original or relevant in the field, and if  then why?  It is a novel issue because in recent years, the rise of social networks and relationships through the Internet have made the problem of bullying also reach through a PC or mobile screen, which young people use diary.  

3. What does it contribute to the subject area compared to other publications?  material?  It contributes to treat important components of cyberbullying such as social skills, self-control of emotions, assertiveness or impulsiveness.  Crucial components to understand cyberbullying from both an emotional and a social point of view.  Going deeper into this problem, to get to understand it better.  

4. What specific improvements could the authors consider regarding the  methodology?  The methodology in this article is correct and the study is clear.  Using material correctly and clearly.

 5. Are the conclusions consistent with the evidence and arguments?  presented and address the main question posed?  Yeah.  The question posed is addressed correctly.  

6. Are the references appropriate?  They are appropriate but some are too old to deal with such a new and current topic.

 7. Include any additional comments about the tables and figures.  Tables are dealt with clearly and directly.  Making the study easier to understand.

Author Response

Reviewer 1

We appreciate your comments. As discussed below, we have revised our manuscript based on your suggestions.

Reviewer’s comment

Current issues are being dealt with and there are references to 1991, which makes no sense. It is not something negative, but a recommendation for the authors to try to update the bibliographical references as much as possible, since there are much more current authors who deal with this topic.

Response:

Thank you for bringing this to our attention. We have reviewed and updated the cited literature by citing newer sources where appropriate and removing older references.

We have removed, among others:

Bandura, A. Social Cognitive Theory of Self-Regulation. Organ. Behav. Hum. Decis. Process. 1991, 50, 248–287, doi:10.1016/0749-5978(91)90022-L

Rotter, J.B. Generalized Expectancies for Internal versus External Control of Reinforcement. Psychol. Monogr. Gen. Appl. 1966, 80, 1–28, doi:10.1037/h0092976

Bandura, A. Social Learning Theory; Prentice-Hall series in social learning theory; Prentice-Hall: Englewood Cliffs, New Jersey, 1977; ISBN 978-0-13-816744-8

- replacing them with:

Spielman, R.M.; Jenkins, W.J.; Lovett, M.D. Psychology 2e; OpenStax: s.l., 2020; ISBN 978-1-951693-23-7.

We have removed,

Cohen, S. Stress, Social Support, and Disorder. In The Meaning and measurement of social support; Veiel, H.O.V., Baumann, U., Eds.; Hemisphere Press: New York, 1992

Widom, C.S. The Cycle of Violence. Science 1989, 244, 160–166, doi:10.1126/science.2704995

- replacing them with:

Prinstein, M.J.; Giletta, M. Future Directions in Peer Relations Research. J. Clin. Child Adolesc. Psychol. 2020, 49, 556–572, doi:10.1080/15374416.2020.1756299.

Wright, K.A.; Turanovic, J.J.; O’Neal, E.N.; Morse, S.J.; Booth, E.T. The Cycle of Violence Revisited: Childhood Victimization, Resilience, and Future Violence. J. Interpers. Violence 2019, 34, 1261–1286, doi:10.1177/0886260516651090.

Reviewer 2 Report

Thank you for your study! It constitutes substantial research. Nevertheless, I suggest the below ameliorations:

1. The objective of the study could be more clearly presented in the abstract

2. Rename the section "Research Problems and hypotheses" as "Research Objective and Hypotheses".

3. Although the background behind the research hypotheses is soundly presented, it would be advisable to clearly define the research hypotheses in a new paragraph following the format:

H1:....

H2:....

4. Please, proofread your manuscript in English to ensure clarity.

1. The study aims at identifying the determinants of cyberbullying to define appropriate intervention. The study performs an in-depth analysis of the phenomenon, focusing on perpetrators in light of a multi-item scale.
2. Although it is not presented, the study endeavors to identify risk factors of cyberbullying using another methodological approach (based on a multi-item scale) rather than building specific risk models or generating prediction models. This is because the study attempts to add to our comprehension of cyberbullying determinants without focusing on controlling cyberbullying. Nevertheless, the regression scheme and the factor analysis outcome are well presented. The research could be used to give an overall picture of cyberbullying determinants before conducting a risk analysis process. Therefore, the research outcome could be used in the initial phase of a cyberbullying risk analysis process.
3. Relevant studies take up the issue of identifying factors of cyberbullying in light of a technical approach based on statistical outcomes, lacking theoretical ground. In contrast to other studies, the research emphasizes the theory behind cyberbullying determinants. In this sense, the research efficiently combines theory with practice.
4. Although the regression scheme and the factor analysis outcome are well presented, the discrimination between preliminary and main results baffles the reader and negatively affects the cohesion of the methodology. Preliminary and main results should be the outcome of a specific analysis. Such analysis could be based on an integrated method. This method could be presented in a sequence of steps or processes.
5. Although the conclusion summarizes the arguments and findings of the research, it doesn't stress the research question because the research hypotheses are not well presented. I suggest that research hypotheses could be better defined to ameliorate the "discussion" section. The discussion "section" attempts to associate results with hypotheses but this association is not strong because the research hypotheses are not clearly defined.
6. The references are appropriate and define the scope of cyberbullying determinants.
7. Although figures and tables are explanatory and illustrate the results appropriately, I suggest that each table and figure should be connected to the part of the results that illustrate and should be renamed respectively (For instance, Table 5 and Figure 3 could be renamed as Table 3.2.1, and Figure 3.2.1 to be connected to the respective part of the results.
8. The manuscript should undergo English proofreading to ensure clarity.

Author Response

Reviewer 2

We appreciate your comments and suggestions. As discussed below, we have revised our manuscript based on most of your suggestions.

Reviewer’s comment

The objective of the study could be more clearly presented in the abstract.

Response:

The study’s main objective was to examine the determinants of being a perpetrator of cyberbullying. We supplemented the description of the study's purpose in the abstract by listing the determinants of cyberbullying that were tested in the study. We hope that the abstract now more clearly presents the main purpose of the study and the range of variables that were examined.

Reviewer’s comment

Rename the section "Research Problems and hypotheses" as "Research Objective and Hypotheses".

Response:

We have made the proposed change.

Reviewer’s comment

Although the background behind the research hypotheses is soundly presented, it would be advisable to clearly define the research hypotheses in a new paragraph following the format:

H1:....

H2:....

[…]

Although the conclusion summarizes the arguments and findings of the research, it doesn't stress the research question because the research hypotheses are not well presented. I suggest that research hypotheses could be better defined to ameliorate the "discussion" section. The discussion "section" attempts to associate results with hypotheses, but this association is not strong because the research hypotheses are not clearly defined.

Response:

We agree that listing the hypotheses and research questions increases the clarity of the purpose and area of study. Due to the length of the text, we have not included a summary of the hypotheses and research questions in a separate, additional paragraph. However, we added the numbers of the hypotheses and questions in brackets in those places in the manuscript where they were defined. This allowed us to refer to them in the discussion of the results. We hope that this allowed for a better association of the results with the hypotheses.

Reviewer’s comment

Please, proofread your manuscript in English to ensure clarity.

Response:

Thank you for pointing this out. The text has been re-checked by a native speaker from a company providing scientific editing services for research articles.

Reviewer’s comment

Although the regression scheme and the factor analysis outcome are well presented, the discrimination between preliminary and main results baffles the reader and negatively affects the cohesion of the methodology. Preliminary and main results should be the outcome of a specific analysis. Such analysis could be based on an integrated method. This method could be presented in a sequence of steps or processes.

Response:

Thank you for pointing out that the division into preliminary and main results may negatively affect the cohesion of the methodology. Therefore, we have dropped the division into preliminary and main results. Instead, we have divided the results section with titles explaining the content of the section. We hope this will increase the readability of the results section and make it easier to return to selected results.

Reviewer’s comment

Although figures and tables are explanatory and illustrate the results appropriately, I suggest that each table and figure should be connected to the part of the results that illustrate and should be renamed respectively (For instance, Table 5 and Figure 3 could be renamed as Table 3.2.1, and Figure 3.2.1 to be connected to the respective part of the results.

Response:

We understand the need to show the link between the results presented in the Table and the Figures. However, the results of all models are shown in a single Table (Table 5), which allows the comparison of results between models. The Figures, due to their volume, must be shown separately. This makes it impossible to unify their numbering. Please note, however, that the title of each Figure begins by indicating the number of the model whose results it presents. In this way, we tried to show the link between the Figures and Table 5.